# Combination-Based Strategies for the Treatment of Actinic Keratoses with Photodynamic Therapy: An Evidence-Based Review

**DOI:** 10.3390/pharmaceutics14081726

**Published:** 2022-08-18

**Authors:** Stefano Piaserico, Roberto Mazzetto, Emma Sartor, Carlotta Bortoletti

**Affiliations:** Dermatology Unit, Department of Medicine, University of Padua, 35121 Padua, Italy

**Keywords:** photodynamic therapy (PDT), actinic keratoses, combination, topical, systemic, laser

## Abstract

Photodynamic therapy (PDT) is a highly effective and widely adopted treatment strategy for many skin diseases, particularly for multiple actinic keratoses (AKs). However, PDT is ineffective in some cases, especially if AKs occur in the acral part of the body. Several methods to improve the efficacy of PDT without significantly increasing the risks of side effects have been proposed. In this study, we reviewed the combination-based PDT treatments described in the literature for treating AKs; both post-treatment and pretreatment were considered including topical (i.e., diclofenac, imiquimod, adapalene, 5-fluorouracil, and calcitriol), systemic (i.e., acitretin, methotrexate, and polypodium leucotomos), and mechanical–physical (i.e., radiofrequency, thermomechanical fractional injury, microneedling, microdermabrasion, and laser) treatment strategies. Topical pretreatments with imiquimod, adapalene, 5-fluorouracil, and calcipotriol were more successful than PDT alone in treating AKs, while the effect of diclofenac gel was less clear. Both mechanical laser treatment with CO_2_ and Er:YAG (Erbium:Yttrium–Aluminum–Garnet) as well as systemic treatment with Polypodium leucotomos were also effective. Different approaches were relatively more effective in particular situations such as in immunosuppressed patients, AKs in the extremities, or thicker AKs. Conclusions: Several studies showed that a combination-based approach enhanced the effectiveness of PDT. However, more studies are needed to further understand the effectiveness of combination therapy in clinical practice and to investigate the role of acitretin, methotrexate, vitamin D, thermomechanical fractional injury, and microdermabrasion in humans.

## 1. Introduction

Actinic keratoses (AKs) are a very common skin disease caused by chronic sun damage. More than 75% of AKs arise in chronically sun-exposed areas such as the face, scalp, neck, and back of the hands/forearms. The rates of malignant transformation vary from 0.025% to 16%, and the risk of progression of squamous cell carcinoma (SCC) increases in patients with multiple AKs [1].

Photodynamic therapy (PDT) is a highly effective and well-tolerated local treatment based on the interaction between a photosensitizer, an appropriate wavelength of light, and oxygen (Figure 1).

It is generally administered with a topical photosensitizer drug, such as 5-aminolevulinic acid (ALA), or derivatives such as methyl aminolevulinate (MAL) [2]. These precursors of the heme biosynthetic pathway are converted within target cells into photoactivatable porphyrins, especially protoporphyrin IX (PpIX). Protoporphyrin IX has its largest absorption peak in the blue region at 410 nm with smaller absorption peaks at 505, 540, 580, and 630 nm^2^. Most light sources for PDT target the 630 nm absorption peak in the red region, associated with greater tissue penetration, although a blue, fluorescent lamp (peak emission 417 nm) is sometimes used with ALA-PDT [3].

The antitumor effects of PDT are based on three principal mechanisms: direct cytotoxic effects on cancer cells, indirect effects due to the fact of tumor vasculature damage, and the activation of the immune response (Figure 1) [4]. PDT with 5-ALA or MAL showed lesion clearance rates of 81–92% for thin and moderately thick AKs on the face and scalp [5]. PDT short-term (3 months) and long-term (12 months) efficacy is comparable or superior to imiquimod 5%, 5-fluorouracil 0.5%, and cryosurgery [6,7,8,9].

The main limitation of PDT compared to other treatments is pain, which may be the reason for stopping the treatment and may induce the patient to decline further treatment. Nerve block, subcutaneous infiltration anesthesia, and cold analgesia are associated with less PDT-related pain [10]. Lower irradiance (as in daylight PDT) and possibly shorter application times are associated with decreased discomfort without loss of PDT efficacy [3].

Nonetheless, PDT is generally considered to give good cosmetic outcomes [6]. This might be of importance when treating lesions localized on the face or the scalp and may explain patients’ preference for and satisfaction with PDT compared to a physically destructive method such as cryotherapy.

However, PDT efficacy may be limited by the reduced ALA/MAL penetration depth and subsequently insufficient PpIX production in lesional skin. Indeed, thicker lesions in some other areas (especially in the upper and lower limbs) are less responsive to this treatment, and some patients show treatment failure or recurrence.

Therefore, AK treatment by combining PDT with other methods of therapy might be more efficacious, although the evidence for such procedures is limited [11].

In this review, we discuss evidence-based in vitro experiments and clinical trials regarding the administration of other therapeutic methods before or after PDT. Specifically, we assessed the effectiveness of topical (i.e., diclofenac, imiquimod, adapalene, 5-fluorouracil, and calcitriol), systemic (i.e., acitretin, methotrexate, vitamin D, and polypodium leucotomos), and mechanical–physical (i.e., radiofrequency, thermomechanical fractional injury ((TMFI)), microdermabrasion, microneedling, and laser) treatment strategies (Figure 2).

## 2. Topical Treatments Combined with PDT

### 2.1. Diclofenac

Diclofenac is a nonsteroidal anti-inflammatory drug that reduces the production of prostaglandins by inhibiting inducible COX-2 (cyclo-oxygenase-2) [12]. It is used for treating AKs, considering that arachidonic acid metabolites promote epithelial tumor growth by stimulating angiogenesis, inhibiting apoptosis, and increasing the invasiveness of tumor cells [12].

Van Der Geer et al. evaluated the effectiveness of 3% diclofenac, applied for four weeks, twice a day, before administering ALA-PDT. The treatment of AKs on the dorsum of the hands of 10 patients had a slightly better long-term outcome (though not significant). Upon evaluating the pain scores during treatment, unbearable pain was detected to a greater extent in the patients of the diclofenac group [13].

However, since 3% diclofenac gel with hyaluronic acid showed better outcomes with longer applications (i.e., 60 and 90 days of administration showed total clearances of 30% and 50%, respectively, in the patients [14]), the effectiveness of PDT treatment in combination with diclofenac and alone should be compared after administration for more than four weeks [15].

### 2.2. Imiquimod

Imiquimod 5% cream stimulates the innate immune response through the interaction with Toll-like receptor 7 (TLR7).

Shaffelburg [16] compared sequential treatment with ALA-PDT versus ALA-PDT alone. Patients with at least 10 actinic keratoses on the face were treated with photodynamic therapy with aminolevulinic acid (20%) at baseline and month 1. At month 2, imiquimod cream (5%) was applied to half of the face, and the vehicle was applied to the other half twice a week for 16 weeks.

At month 12, the median reduction in the lesion was 89.9% and 74.5% (*p* = 0.0023) on the parts of the face treated with the combination of the imiquimod cream with ALA-PDT and ALA-PDT alone, respectively. Furthermore, tolerance was similar in patients treated with imiquimod and those treated with placebo. Serra et al. [17] compared the effectiveness of PDT alone, imiquimod (5%) alone, and the combined administration in 105 patients (PDT followed by imiquimod twice a week for 16 weeks).

They reported a complete clinic-pathologic response (*p* = 0.038) in 10% of the patients in the PDT-alone group, 27% of the patients in the imiquimod-alone group, and 34% of the patients in the PDT + imiquimod group. Moreover, a significantly greater percentage of patients who received PDT were extremely satisfied with the treatment than those who received only imiquimod (90% PDT vs. 61% imiquimod). Another study with an open-label parallel cohort design investigated the effectiveness of the treatment with imiquimod cream (3.75%), combined with photodynamic therapy (ALA-PDT) vs. imiquimod cream (3.75%) alone [18]. The former group received imiquimod (3.75%) daily for two weeks in a two-week cycle, separated by two weeks of no treatment and followed by one session of photodynamic therapy of the entire face with ALA and blue light. In the latter group, the treatment was administered daily for two weeks in a two-week cycle, separated by two weeks of no treatment. A mean reduction rate of 81% was found for the group treated with PDT, and 83% for the group treated only with imiquimod (the mean difference between the combination treatment and treatment with only imiquimod cream (3.75%) was 2%). However, five out of nine patients who received the combination treatment showed complete clearance of AKs, while only two out of nine patients showed this outcome in the other group. The rates of adverse events observed between the two groups were similar.

### 2.3. Topical Retinoids

Retinoid drugs are also used widely for the treatment of AKs, as they can disrupt gene expression, modify cell differentiation, and modulate cell proliferation or hyperplasia [19].

Currently, three generations of synthetic retinoids exist. First-generation retinoids include tretinoin (all-trans RA), isotretinoin (13-cis-retinoic acid), and alitretinoin (9-cis RA). Second-generation retinoids include etretinate and acitretin. Third-generation retinoids include adapalene, tazarotene, and bexarotene [1,2]. The application of tazarotene gel (0.1%) twice daily on AKs of the upper extremities one week before the administration of ALA-PDT (20%) increased the reduction in the lesion count by ≥50% (*p* = 0.0547) [20].

Galitzer [21] studied the effectiveness of adapalene gel (0.1%) as a pretreatment topical drug for the treatment of AKs on the dorsal hands and forearms by administering a single dose of ALA-PDT. A 10% ALA gel was used instead of a 20% ALA gel, and a red light was used for illumination instead of blue light. The difference between the two groups was significant (*p* = 0.0164), with a median lesion count reduction of 79% in the adapalene pretreated group compared to 57% in the standard therapy group. The discomfort under red light was slightly greater in patients of the standard therapy group, but the difference was not significant.

### 2.4. 5-Fluorouracil Cream

Topical 5-fluorouracil (5-FU) has greater efficacy than ALA and MAL-PDT [22].

It is converted in the cell to a thymidylate synthase inhibitor that interrupts the synthesis of thymidine, causing growth arrest and apoptosis [23].

In mouse models of SCC (SKH-1 and A431 mice), pretreatment with 5-FU for three days followed by ALA for 4 h caused a considerable, tumor-selective increase in PpIX levels and enhanced cell death upon illumination. The effect might be related to a change in the expression of the heme enzyme (upregulating coproporphyrinogen oxidase and downregulating ferrochelatase) and to a substantial, tumor-selective increase in apoptosis [24].

Likewise, an increase in the PpIX levels (two-fold to three-fold) in 5-FU-pretreated lesions versus controls was also reported in patients with AKs [25].

The clearance rates after treatment with MAL-PDT, with or without six days of 5-FU pretreatment, were 75% and 45% at three months and 67% and 39% at six months, respectively [26].

Niessen et al. [27] found that pretreatment with 5-FU for a week, twice daily before treatment with daylight (DL)-PDT, was more effective for AKs on the dorsal side of hands compared to treatment with DL-PDT alone. At the three-month follow-up, the overall lesion response rate was significantly higher for the combination treatment (62.7%) than for the treatment with DL-PDT alone (51.8%), while no difference was found in the degree of erythema one day after PDT between the treatment groups (pretreatment with 5-FU only caused minimal erythema in a few patients before illumination).

Other studies confirmed that the combined treatment was more effective for AKs on the upper limbs or the face [28,29,30].

### 2.5. Topical Calcitriol/Calcipotriol

Vitamin D is a differentiation-promoting hormone, and calcitriol is the most potent and active form of the hormone, the receptor of which is also present in cancer cells [22].

Furthermore, it induces a marked increase in protoporphyrin IX accumulation at the site of the tumor and promotes the activation of the extrinsic apoptotic pathway [31]. Therefore, cell damage after light irradiation is considerably higher in calcitriol-pretreated cells [32,33,34].

Galimberti first proposed pretreatment with calcipotriol (50 mcg/g) during the treatment of actinic keratoses with MAL DL-PDT [35]. Three months after the treatment, the complete response rate was 85% after calcipotriol pretreatment, while it was 70% for the DL-PDT alone group. Calcipotriol/DL-PDT showed a stronger association with erythema than DL-PDT alone. Some patients preferred DL-PDT alone due to the inconvenience caused by the daily application of calcipotriol and the related erythema and desquamation.

Torezan et al. confirmed these findings and showed improvement in keratinocyte atypia following treatment with MAL-PDT, with slightly more improvement on the parts pretreated with calcipotriol. However, erythema, edema, crusting, and postulation represented the most common adverse events and occurred more frequently in the group treated with the topical vitamin D3 analog [36].

Moreover, a long-term study on treatment with MAL-PDT with prior application of topical calcipotriol or conventional MAL-PDT for AKs on the scalp showed an overall AK clearance of 92% and 82% after three months for CAL-PDT and conventional PDT, respectively (*p* < 0.001). Similar results were found at 6 and 12 months, i.e., 92% vs. 81% (*p* < 0.001) and 90% vs. 77% (*p* < 0.001) for CAL-PDT and conventional PDT, respectively [37].

Another intra-individual, randomized trial [38] was conducted with patients with AKs on the dorsum of the hands or forearms, who were treated with another vitamin D analog, calcitriol (3 mg/g). Calcitriol or a placebo was applied once daily for 14 consecutive days. On day 15, the first administration of MAL DL-PDT was performed, and the second one took place after one week. A higher efficacy was found for grade II and grade III AKs, with a response rate of 55.24% for the group pretreated with calcitriol and 39.58% for the control group (*p* = 0.038).

The administration of topical vitamin D analogs might be more effective than treatment with PDT alone, particularly for thicker and difficult-to-treat AKs on the upper extremities and the scalp. However, the administration of vitamin D analogs might be associated with an increase in local skin reactions.

## 3. Systemic Treatment Combined with PDT

### 3.1. Acitretin

Acitretin is a systemic retinoid drug. Although the on-label instruction for acitretin only recommend its use for psoriasis, its off-label use in keratinization disorders is very common [39].

Acitretin induces normalization of differentiation and proliferation as well as the modification of inflammatory responses and neutrophil function in the skin. Moreover, some studies have shown the effectiveness of this drug in preventing non-melanoma skin tumors and AKs [40].

In a nonclinical setting, retinoic acid preconditioning before PDT can induce a moderate (though nonsignificant) increase in ALA–PpIX levels in prostate cancer cells. However, Hasan et al. found a dramatic increase in PDT response after pretreatment with retinoids due to the increase in the production of protoporphyrin IX (PpIX) in the target cells [41,42].

Preclinical studies support the hypothesis that induction of keratinocyte differentiation can increase intracellular PpIX accumulation in cells treated with ALA before light exposure in mice [41]. Treatment with a combination of acitretin and ALA-PDT can significantly increase the apoptosis rate and the mortality rate of SCL-1 cells compared to treatment with acitretin or ALA-PDT alone [43].

However, we found no studies regarding the effectiveness of combining PDT with acitretin in clinical trials or observational studies. This might be an important treatment strategy, considering the encouraging results reported in vitro.

### 3.2. Methotrexate

Methotrexate (MTX) is widely used for the treatment of psoriasis and other conditions characterized by cell hyperproliferation and lack of differentiation. MTX is a chemotherapeutic agent that inhibits cell proliferation and triggers cellular differentiation [44].

Pretreating human SCC13 cells, HEK1 cells, and normal keratinocytes with MTX for 72 h enhanced the PpIX levels by two-fold to four-fold in cancerous cells relative to the PpIX levels in nontumoral keratinocytes [45]. This was probably because MTX modified the expression of certain enzymes involved in the porphyrin metabolic pathway (a four-fold increase in coproporphyrinogen oxidase and a stable or slight decrease in the expression of ferrochelatase) and induced the expression of differentiation markers (i.e., E-cadherin, involucrin, and filaggrin) in cancer cells [45]. Photodynamic cell killing was thus synergistically enhanced by the combined therapy compared to the effectiveness of PDT alone.

However, the oral administration of MTX before PDT in patients with AKs, specifically in the elderly and those with several comorbidities, might not be possible, since the risks could exceed the benefits compared to the risks with other topical or systemic drugs.

### 3.3. Vitamin D

We previously discussed the ability of vitamin D derivatives and their prodifferentiating effects to increase protoporphyrin IX accumulation and enhance ALA-PDT-mediated cell death.

Systemic delivery of calcitriol is easily performed in mice and increases the tumoral accumulation of PpIX up to 10-fold, while in humans, high calcitriol levels are associated with the risk of developing hypercalcemia. Interestingly, the deficiency of serum vitamin D is associated with poorer responses of AKs to MAL-PDT [46] and, thus, further studies are needed to investigate its role in the treatment of AKs.

### 3.4. Polypodium Leucotomos

Polypodium leucotomos (PL) is an effective systemic photoprotective agent that strongly protects the skin against UV radiation [47].

PL is extracted from ferns grown in Central America and has been used for centuries by Native Americans for treating malignant tumors. Studies have shown that PL has antioxidative properties, immunomodulatory properties, and antitumoral activity [48].

Specifically, PL extract supplementation induces p53 activation and reduces acute inflammation by inhibiting the Cox-2 enzyme and increasing the removal of cyclobutane pyrimidine dimers [49,50].

In clinical settings, PL supplementation at a dose of 960 mg per day for one month and then 480 mg per day for five months decreased the recurrence rate of AKs at six months after two sessions of MAL-PDT, administered one week apart. MAL-PDT + PL supplementation showed a median reduction of 87.5% in scalp AKs compared to a reduction of 62.5% in the group treated only with MAL-PDT (*p* = 0.040). Moreover, no major side effects were recorded in either group [51].

Therefore, using PDT-MAL combined with PL might be an effective treatment for AKs due to the fact of its effect on reducing immunosuppression in the human skin not only after UV irradiation but also after PDT treatment, thus reducing the recurrence of AKs in high-risk individuals.

## 4. Physical and Mechanical Treatments Combined with PDT

Most of the aforementioned topical and systemic drugs improve PDT efficacy by preferentially enhancing PpIX production in AK cells, acting as differentiation-promoting agents.

However, PDT is mostly limited by the poor ALA/MAL penetration within lesional skin. The skin barrier is a lipid layer along which molecules can migrate through diffusion; they can enter through transcellular pathways, intercellular spaces, and the trans-appendageal pathway that includes hair follicles, sebaceous glands, and sweat glands [52,53].

One extensively investigated strategy to enhance drug delivery through the skin is by adding chemical enhancers such as fatty acids, surfactants, esters, and alcohols [54,55].

However, mechanical, physical, and active transport techniques also enhance skin penetration [56]. Some of these treatment methods have been assessed in combination with PDT for AK treatment. The disruption and ablation of the stratum corneum (the primary barrier to the delivery of topical drugs) by using lasers, radiofrequency (RF), thermomechanical fractional injury (TMFI), microdermabrasion (MD), or microneedling can enhance PDT treatment.

### 4.1. Laser-Assisted PDT

Different types of light sources are used for topical PDT including fluorescent lamps, light-emitting diodes (LEDs), and lasers [2]. However, lasers are not widely used because of their side effects, including pain and discomfort during treatment, as well as erythema, blistering, crusting, and pigmentation after treatment. Moreover, their application is limited while treating larger fields [57].

Due to the advancement in technology, fractional ablative lasers have been developed for promoting topical/transdermal drug delivery. Er:YAG (erbium:yttrium–aluminum–garnet) and CO_2_ (carbon dioxide) lasers produce microscopic vertical channels in the skin that enhance the penetration of photosensitizers [58].

Thus, lasers can be used during pretreatment to facilitate the enrichment of ALA or MAL in dysplastic cells, and this approach is also known as laser-assisted drug delivery.

#### 4.1.1. Er:YAG (Erbium:Yttrium–Aluminium–Garnet) Lasers

Shen et al. studied the in vivo kinetics of protoporphyrin IX (PpIX) after topical ALA application, enhanced by an Er:YAG laser; the increase in the ratio of PpIX with laser-treated murine skin ranged from 1.7 to 4.9-times compared to the PpIX levels in the control group [59].

Gellén et al. [60] conducted an intra-patient randomized study and showed that fractional laser pretreatment leads to significantly higher clearance at three months compared to the clearance with PDT alone. However, no difference in recurrence rates was found during follow up after 12 months. In every patient, two random treatment areas received conventional PDT or Er:YAG-ablative fractional laser-assisted PDT (AFL-PT). The number of AKs decreased by 87.5 ± 17.3% and 82.5 ± 16.5% (*p* = 0.039) three months after Er:YAG-AFL PDT and cPDT, respectively. Furthermore, upon assessing the immune infiltration, a reduction in Ki67-positive cells and CD8+ T cells was found three months after Er:YAG-assisted PDT [60].

Togsverd-Bo et al. [61] examined difficult-to-treat AKs in organ-transplant recipients (OTRs) using fractional Er:YAG laser-assisted DL-PDT. At three months, the clearance rates were 74% after AFL-dPDT, 46% after dPDT, 50% after cPDT, and 5% after AFL (*p* < 0.001). AFL-dPDT had excellent tolerability, even if the pain was higher in areas treated with AFL-assisted DL-PDT than in areas treated with DL-PDT only.

Another prospective randomized nonblinded trial [62] was conducted to evaluate patients who underwent one session of MAL-PDT using a red light-emitting diode lamp at 37 J/cm^2^, and the patients were randomly assigned to receive fractional Er:YAG laser (AFL-PDT) pretreatment. AFL-PDT was significantly more effective than MAL-PDT for treating patients with AKs of all grades (86.9% vs. 61.2%; *p* < 0.001), although the efficacy of AFL-PDT was the most pronounced in treating Olsen grade III AKs (69.4% vs. 32.5%; *p* = 0.001). AFL-PDT also showed a lower lesion recurrence rate than MAL-PDT (9.7% vs. 26.6%; *p* = 0.004), and an excellent or good cosmetic outcome was reported in >90% of the cases. Erythema and hyperpigmentation intensities were higher but not significant in the AFL-PDT group, while the side effects were mild but more frequent in the AFL-PDT group, although the result was not statistically significant [62].

#### 4.1.2. CO_2_ (Carbon Dioxide) Laser

Haedersdal et al. evaluated drug delivery of ALA and MAL using CO_2_ ablative fractional laser (AFXL). They found that AFXL increased topical uptake of porphyrin precursors when it was followed by the topical application of MAL for 3 h. This pretreatment with laser generated 3 mm microchannels in the skin and, consequently, enabled a more homogeneous distribution of the porphyrins throughout the skin [63].

Syed et al. [64] studied 12 OTRs with multiple (>5) AKs of identical size in two areas on the scalp and/or the forehead. The treatment areas, randomized to AFL-assisted DL-PDT, were first treated with a carbon dioxide laser (eCO_2_) targeting single AK lesions, followed by treatment of the whole field with the methyl aminolevulinate (MAL) cream applied to both treatment areas. After 30 min, both treatment areas were exposed to sunlight for 2 h. At the four-month follow up, the overall complete response was 75.5% in areas treated with AFL-assisted DL-PDT and 64.0% in areas treated with DL-PDT. There was a significant interaction between a complete response and the AKs’ grade (*p* = 0.001), as well as between a partial response and the AKs grade (*p* = 0.007). Patient-reported pain was significantly higher in areas treated with AFL-assisted PDT in the first two days (*p* = 0.008) but not after five days (*p* = 0.11) [64].

Togsved-Bo et al. [65] found that at the three-month follow up, AFL-PDT was significantly more effective than PDT for all grades of AKs. Complete lesion response of grade II-III AKs was 88% after AFL-PDT and 59% after PDT (*p* = 0.02). In grade I AKs, 100% of the lesions were cleared after AFL-PDT, while 80% of the lesions were cleared after PDT (*p* = 0.04). The AFXL-PDT-treated skin showed significantly fewer new AK lesions (*p* = 0.04) and better photoaging (*p* = 0.007) than the PDT-treated skin. The pain scores during illumination (*p* = 0.02), erythema, and crusting were higher. Inflammatory reactions were more intense in AFL-PDT-treated skin than in PDT-treated skin in 83% of the patients and were of equal intensity in 17% of the patients. PpIX fluorescence was higher in AFL-pretreated skin (*p* = 0.003) [65] and supported the clinical superiority of the combination treatment.

Paasch et al. evaluated the effectiveness and safety of CO_2_ AFL-LAD combined with indoor daylight (IDL) ALA-PDT for treating skin field cancerization associated with AKs. All patients showed remission (complete: 71.7%; partial: 28.3%), suggesting that AFL-LAD combined with IDL-PDT is an exceptionally effective treatment approach. However, the high pain score associated with this combined approach is a limiting factor [66].

Jang et al. [67] showed that laser-assisted PDT might reduce the photosensitizer incubation time or the number of sessions required for complete response (efficacy of 70.6% in three sessions).

In conclusion, several studies showed a reduction in the number of AKs and an improvement in the recurrence rate in the approach combining laser treatment and PDT. A recent meta-analysis [30] reported an overall significantly high clearance rates for laser-assisted PDT than for PDT alone (RR: 1.33, 95% CI: 1.24–1.42); however, the evidence for this outcome was graded as low (GRADE ++--). Furthermore, the pain was similar for both treatments (mean difference of 0.31, 95% CI: 0.12–0.74, low quality of evidence, GRADE ++--).

### 4.2. Radiofrequency and Thermomechanical Fractional Injury (TMFI)

Fractional radiofrequency (RF) creates plasma close to the skin and provokes plasma sparks on the skin’s surface that induce epidermal ablation and produce microchannels that perforate the dermis, thus improving drug delivery. Additionally, sonophoresis facilitates the movement of molecules through the intact skin under the influence of an ultrasonic perturbation. Low-frequency ultrasound (frequencies below 100 kHz) can highly enhance transdermal transport [68].

Park et al. demonstrated that fractional RF with sonophoresis effectively enhanced ALA penetration in swine skin [69]. Prefractional RF followed by post-treatment with sonophoresis is a promising therapeutic combination for ALA-PDT to enhance ALA uptake [67]. However, no studies on humans have been conducted yet.

Shavit et al. [70] evaluated the efficacy of pretreatment by thermomechanical fractional injury (TMFI) at low-energy settings in five healthy subjects to increase the permeability of the skin to four topical preparations. TMFI, (Tixel^®^;Novoxel^®^, Netanya, Israel), is a thermomechanical system developed for providing fractional treatment. The system is designed for treating soft tissues by direct conduction of heat, which allows water to evaporate rapidly while causing low thermal damage to the surrounding tissue.

The authors evaluated the permeability of 20% ALA gel, 10% ALA microemulsion gel, 16.8% MAL cream, and 20% ALA hydroalcoholic solution. Pretreatment with low-energy TMFI at a pulse duration of 6 milliseconds increased the percutaneous permeation of ALA when the 20% gel was used.

Interestingly, after 2 h and 3 h of treatment, the TMFI-treated sites exhibited a higher hourly rate of PpIX fluorescence intensity, which was 156–176% higher than that in the control (*p* ≤ 0.004). Thus, TMFI is a powerful method to enhance the transdermal drug delivery of ALA and its derivatives. Further studies are needed to investigate its role in improving PDT treatment.

### 4.3. Microdermabrasion (MD)

Microdermabrasion (MD) is an easily accessible and cost-effective technique that can be used to increase the efficacy of PDT as a pretreatment method.

Bay et al. compared PpIX accumulation after a range of standardized pretreatments and found significantly higher median intracutaneous PpIX fluorescence intensities after AFL compared to that with alternative physical modalities such as MD [71].

The clinical efficacy of PDT was evaluated after pretreatment with either Er:YAG AFL or MD in a side-by-side trial [72].

MD pretreatment was performed using a pad with particles that had a diameter of 58.5 µm (Ambu^®^ Skin Prep Pads 2121M; Ambu A/S, Ballerup, Denmark). Two large areas were randomly selected in each individual, and a single treatment was administered with AFL + DL-PDT or microdermabrasion + DL-PDT. Interestingly, AFL-dPDT resulted in a significantly higher AK clearance (81% vs. 60%, *p* < 0.001), led to fewer new AKs (*p* < 0.001), and showed higher improvement in dyspigmentation (*p* = 0.003) and skin texture (*p* = 0.001), which was related to microneedling-assisted DL-PDT (MD-dPDT). However, laser-assisted PDT caused more local skin reactions than microdermabrasion pretreatment [72].

### 4.4. Microneedling-Assisted PDT

Microneedling represents another pretreatment option to improve the effects of PDT. A microneedle consists of tiny needles on a roller or a stamp that punctures the skin and forms microchannels.

In vivo experiments using nude mice showed that microneedle punctures could reduce the application time and ALA dose required to induce high levels of the photosensitizer protoporphyrin IX in the skin. This is beneficial for clinical practice, as shorter application times are more convenient for patients and clinicians [73].

The Microneedling Photodynamic Therapy II (MNPDT-II) study revealed a statistically significant difference in the clearance of AKs for the 20 min incubation arm between the microneedling side and the PDT alone side (76% and 58%, respectively). The microneedle device consisted of a single-use sterile array of microneedles (200 µm long), while the sham treatment consisted of the applicator roller without microneedles. Immediately after sham and microneedle pretreatment, topical ALA was applied to the entire face, followed by exposure to blue-light PDT [74].

Likewise, Spencer and Freeman reported that microneedling pretreatment induced a significantly greater reduction in the AK count compared to that after PDT alone (89.3% and 69.5%, respectively); the occurrence of side effects was similar [75].

Furthermore, a comparative study conducted by Chen et al. to evaluate ALA-PDT revealed that plum-blossom needling (a method of shallow insertion of multiple needles into the skin) caused a broader diffusion of ALA than that with CO_2_ AFXL, and the technique had a similar clinical effect at a considerably lower cost. The needle-pretreated-lesion had a stronger surface fluorescence intensity than the laser-pretreated-lesion [76].

However, Lev-Tov et al. [77] conducted a randomized controlled evaluator-blind trial and found no significant differences in the efficacy between microneedling-PDT and ALA-PDT (with a short incubation time of 60 min), while the microneedling side showed significantly higher pain scores. The authors stated that this conflicting result was due to the use of shorter microneedles, which only reached the epidermis (and not the dermis), than those used in other studies.

## 5. Other Physical and Chemical Treatments

Other methods have been used to increase the effectiveness of PDT. However, no studies have compared them to conventional-PDT or to daylight-PDT for the treatment of AKs.

In vivo studies showed that the incorporation of a thermogenic and vasodilating substances, such as methyl nicotinate, in the MAL cream increased the amount of PpIX produced in the tissue, causing a greater effect on the epidermis after PDT as well as reducing the cream’s incubation time [78].

Iontophoresis, a physical treatment aimed at facilitating transport through the skin of ionic species using a voltage-gradient applied to an electrolytic formulation [23], was reported to reduce the incubation time by 1 h in a study comparing iontophoresis-assisted AFL-PDT with AFL-PDT [79].

Photobiomodulation consists of the illumination of tissues with subthermal radiometric conditions (red or near-infrared), stimulating cell metabolism and enhancing the production of PpIX and the effectiveness of PDT in vitro and in vivo tumors (human glioma cells) [80].

Another possible approach to increase 5-ALA penetration and efficiency is the addition of glycolic acid (GA) to 5-ALA for the treatment of patients with SCC [81] and ethylenendiamine-tetra-acetic-acid (EDTA) and dimethylsulphoxid (DMSO) to 5-ALA in order to improve protoporphyrin IX accumulation in certain subtypes of BCC, SCC, and Bowen’s disease [82].

Therefore, iontophoresis, photobiomodulation, and the addition of methyl nicotinate, GA, or DMSO in ALA-based topicals represent encouraging possibilities to enhance conventional-PDT or daylight-PDT efficacy and should be further studied to assess their role in treating AKs as well.

## 6. Conclusions

Our review showed that combination treatment may significantly improve the effectiveness of PDT in reducing AKs (Table 1). Pretreatment with imiquimod, adapalene, 5-FU, and calcipotriol showed greater effectiveness in treating AKs than PDT alone, while evidence supporting the effectiveness of microneedling and diclofenac gel was poor.

Pretreatment with 5-FU, calcitriol, and adapalene can substantially increase the effectiveness of PDT in the extremities (10.9%, 15.7%, and 22% more than PDT alone, respectively). Immunosuppressed patients may benefit from the administration of 5-FU cream pre-PDT or by the combination of PDT with laser. In particular, Er:YAG laser and CO_2_ laser-assisted PDT showed greater efficacy than PDT alone (28% and 11.5%, respectively). Treatment of thicker AKs with PDT can be improved by combination with Er:YAG laser-PDT (69.4% effectiveness, 36.9% more than PDT) even though a slight increase in the incidence of side effects was found and should be considered. AKs of higher grade may also be pretreated with higher effectiveness with CO_2_ laser-PDT or with calcitriol (3 mg/g).

Therefore, combining treatments may greatly enhance PDT’s ability to reduce AKs. However, more studies are needed to further understand the effectiveness of combination therapy in clinical practice and to investigate the role of systemic drugs, such as acitretin, methotrexate, and vitamin D, in humans.

## Figures and Tables

**Figure 1 pharmaceutics-14-01726-f001:**
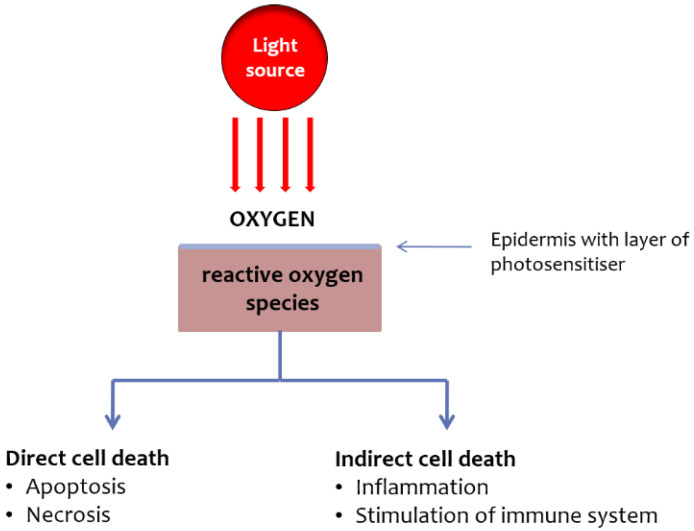
Schematic representation of PDT’s mechanism of action.

**Figure 2 pharmaceutics-14-01726-f002:**
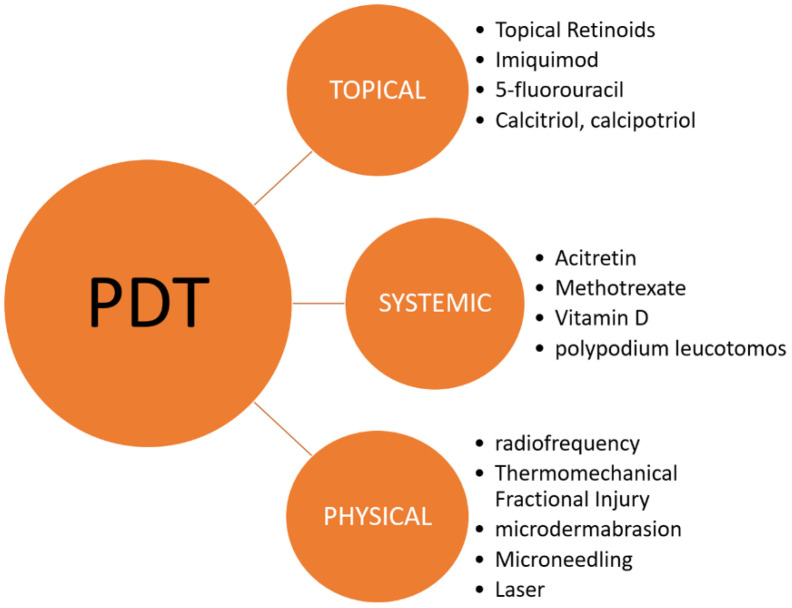
Combination-based strategies for the treatment of actinic keratoses with PDT.

**Table 1 pharmaceutics-14-01726-t001:** A list of comparative studies regarding the treatment of AKs with PDT alone and combination-based PDT in the general population.

Combining Therapy	Regimen Adopted	Outcome of Combined Regimen vs. PDT Alone	Adverse Effects of Combined Regimen vs. PDT Alone	Reference
**TOPICAL TREATMENTS**				
**DICLOFENAC 3% gel** pKa = 4.2 * logP = 0.7 **	Twice daily for 4 weeks, then one session with ALA-PDT.	12 month decrease in the total number of lesions score of 12.5 in the diclofenac group, while it was 8.8 in the control group. Not significant (*p* = 0.34).	When looking at the pain scores during treatment, a tendency for a greater, unbearable pain was scored in the diclofenac group.	[13]
**IMIQUIMOD 5%** pKa = 19.99 ^§^ logP = 2.65 ^§^	Treated with ALA-PDT followed by imiquimod (3 times a week for 4 weeks).	Complete clinicopathologic response (*p* = 0.038) was obtained in 10% of the PDT-alone group, 27% in imiquimod-alone group, and in 34% of the PDT + imiquimod group.	No significant differences were observed among the options and tolerance to treatment.	[17]
	Treated with ALA-PDT at baseline and at month 1. At month 2, imiquimod 5% cream was applied 2 times per week for 16 weeks.	Median lesion reductions were 89.9% versus 74.5% (*p* = 0.0023), respectively, with a median difference of combination vs. ALA-PDT of 15.5%.	Similar.	[18]
**TAZAROTENE 0.1%** pKa = 1.23 ^§^ logP = 4.2 °	TZ gel 0.1% twice daily on AKs of the upper extremities, 1 week before ALA-PDT with ALA 20% gel.	Lesion count reduction ≥ 50% eight weeks after. The significance was borderline (*p* = 0.0547).	Adverse events were limited to those expected after ALA-PDT. In the pretreated arm five minutes after ALA-PDT, erythema was significantly more severe (*p* = 0.0029).	[20]
**ADAPALENE 0.1%** pKa = 3.99 ^§^ logP = 6.46 ^§^	Adapalene 0.1% gel twice daily for one week, then one session with ALA-PDT with ALA 10%.	A median lesion count reduction in the adapalene-pretreated group of 79% compared to 57% in the standard therapy group, with a median difference of 22%. (*p* = 0.0164)	Discomfort during PDT was slightly greater with the standard therapy, but the difference did not achieve significance.	[21]
**5-FLUOROURACIL 5%** pKa = 8.02 ^φ^ logP = 0.89 **	5-FU 5% cream + MAL-PDT. Pretreatment with 5-FU 5% cream for 6 days followed by one session of MAL-PDT. AKs of the face, scalp, and forearms.	Relative clearance rates after PDT with or without 5-FU pretreatment were, respectively, 75% versus 45% at 3 months (mean difference of combinations was 30%) and 67% versus 39% at 6 months (mean difference of combinations was 28%).	5-FU/PDT combination treatment was well tolerated, with no major side effects other than the local inflammatory reaction typically associated with PDT treatment.	[25]
	5-FU 5% cream + dl-PDT. Pretreatment with 5-FU 5% cream twice daily for 7 days followed by one session of dl-PDT. AKs on the dorsal side of hands.	The reduction rate (mean) of the combined treatment group was 62.7%, while it was 51.8% in the PDT-alone group. The difference in combinations vs. monotherapy (mean) was 10.9% (*p* = 0.001).	No difference was found in the degree of erythema one day after PDT between the 2 treatment groups.	[27]
	Pretreatment with 5-FU 5% cream twice daily for 7 days followed by one session of ALA-PDT. AKs of the face.	A median lesion count reduction in the 5-FU-pretreated group of 100% compared to 66.7% in the standard therapy group with a median difference of 33.5%.	No significant difference in discomfort.	[29]
	5-FU 5% cream twice daily for 7 days followed by one session of ALA-PDT. Unclear regarding the localization of the KAs.	A median lesion count reduction in the 5-FU-pretreated group of 94.6% compared to 68.4% in the standard therapy group, with a median difference of 26.2% (*p* = 0.001).	Similar.	[30]
**CALCIPOTRIOL** **50 mcg/g** pKa = 14.39 ° logP = 3.84 °	15 days of treatment with calcipotriol or placebo (once daily) followed by one session of MAL-daylight-PDT.	The complete response rate was 85% while it was 70% for the dl-PDT-alone group; the partial response rate was 12% and 25%, respectively.	Calcipotriol/DL-PDT was associated with more marked erythema than that observed with DL-PDT alone.	[35]
	Calcipotriol was applied daily for 15 days beforehand on the other side.	At three months, overall AK clearance was 92.07% and 82.04% for CAL-PDT and conventional PDT, respectively (*p* < 0.001). Similar results were found at 6 and 12 months: 92.07% and 81.69% (*p* < 0.001), and 90.69% and 77.46% (*p* < 0.001) for CAL-PDT and conventional PDT, respectively.	Slightly superior discomfort after the application of calcipotriol.	[36,37]
**CALCITRIOL** **3 mg/g** pKa = 14.39 ° logP = 4.35 °	A layer of calcitriol 3 mg/g or placebo was applied once daily for 14 consecutive days. On day 15 first MAL-DL-PDT was performed, while the second one took place 1 week apart.	A higher efficacy was found for the grade II and grade III AK groups. The response rate was 55.24% for the group pretreated with calcipotriol, whereas it was 39.58% for the control group, with a difference of 15.66% (*p* = 0.038).	Local skin reactions occurred more frequently on the calcitriol DL-PDT-treated sides.	[38]
**SYSTEMIC TREATMENT**				
**POLYPODIUM** **LEUCOTOMOS**	One week after MAL-PDT, PLE supplementation at a dose of 960 mg per day for 1 month and then 480 mg per day for 5 months.	At the 6 month follow up, PDT treatment + PLE supplementation displayed a better clearance rate compared with PDT alone (*p* = 0.040). There was a median reduction in scalp AKs of 87.5% in the combination group, while that of 62.5% in the group treated just with PDT.	No major side effects were recorded in either group.	[51]
**PHYSICAL AND MECHANICAL TREATMENT COMBINED WITH PDT**				
**Er:YAG** **ABLATIVE FRACTIONAL LASER-ASSISTED PDT (AFL-PT)**	The side affected was pretreated with Er:YAG-AFL immediately before ALA application, then PDT was undertaken with 20% ALA, applied for 3 h, then irradiated with water-filtered infrared A light for 20 min.	The number of AKs decreased by 87.56 ± 17.30% and 82.56 ± 16.53% (*p* = 0.039) 3 months after Er:YAG-AFL PDT and cPDT, respectively.	Not reported.	[60]
	The side affected was pretreated with Er:YAG-AFL, immediately before MAL application, then PDT was undertaken with a red-light-emitting diode (LED) lamp.	FL-PDT was significantly more effective than MAL-PDT at treating all AK grades (86.9% vs. 61.2%; *p* < 0.001). The efficacy of FL-PDT was most pronounced in treating Olsen grade III AKs (69.4% vs. 32.5%; *p* = 0.001). FL-PDT also showed a lower lesion recurrence rate than MAL-PDT (9.7% vs. 26.6%; *p* = 0.004).	Erythema and hyperpigmentation intensities were higher but not significant in the FL-PDT group, while side effects were mild but more frequent in the FL-PDT group, even though this result was not statistically significant (*p* > 0.05).	[61]
**CARBON DIOXIDE LASER (ECO_2_)-ASSISTED PDT** **(AFL-PT)**	Carbon dioxide laser (eCO_2_), first targeting single AK lesions, followed by treatment of the whole field with methyl aminolaevulinate (MAL) cream applied on both treatment areas. Red-light PDT was used.	At 3 months follow up, the complete lesion response of grade II–III AKs was 88% after AFXL-PDT compared with 59% after PDT (*p* = 0.02). In grade I AKs, 100% of the lesions cleared after AFXL-PDT compared with 80% after PDT (*p* = 0.04). AFXLPDT-treated skin responded with significantly fewer new AK lesions (*p* = 0.04).	Pain during LED illumination was significantly higher in AFXL-PDT-treated areas than in PDT-treated areas. After treatment, patients developed erythema and crusting in both treatment areas.	[65]
	Carbon dioxide laser (eCO_2_), first targeting single AK lesions, followed by treatment of the whole field with ALA cream. Red-light PDT was used.	After the study protocol, all patients showed remission (complete: 71.7%; partial: 28.3%).	Higher pain scores were associated with this combined approach.	[66]
	Carbon dioxide laser (eCO_2_), first targeting single AK lesions, followed by treatment of the whole field with 20% ALA or MAL. Red-light PDT was used.	70.6% of the lesions showed a complete response (CR) within three sessions of PDT.	No significant side effects were associated with the combination of ablative CO_2_ fractional laser and PDT.	[67]
**MICRONEEDLE-ASSISTED PDT**	The microneedle device consisted of a single-use sterile array of microneedles 200 µm in length. Immediately after microneedle pretreatment, each topical ALA was applied to the entire face, and blue-light PDT was used.	Participants experienced significantly superior AK lesion clearance (76% vs. 58%, *p* < 0.01) at 20 min incubation times. While the 10 min group also experienced improvement in AK counts, the clearance rates between the microneedle side and the sham side were not significantly different.	The secondary outcome of pain associated with blue-light exposure during PDT was nominal and not significantly different from the sham side.	[74]
	Microneedling device applied to ½ of their face was followed by applying ALA 20% cream. Subsequently, blue-light PDT was used.	The mean percentage reduction in AKs was 89.3% on the microneedling side versus 69.5% on the PDT-alone side. There was a significant difference.	Not different.	[75]
	Microneedling device applied to ½ of their face was followed by applying ALA 20% cream for 60 min incubation. Subsequently, blue-light PDT was used.	The average complete response rates for 20, 40, and 60 min microneedling times versus ALA-PDT were 71.4% and 68.3%; 81.1% and 79.9%; 72.1% and 74.2%, respectively. There were no statistically significant differences.	There was statistical significance in pain scores between the microneedling application and the control one, but the absolute difference was small.	[77]

* Settimo, L.; Bellman, K.; Knegte, R.M.A. Comparison of the Accuracy of Experimental and Predicted pKa Values of Basic and Acidic Compounds. *Pharm. Res*. 2014, 31, 1082–1095; https://doi.org/10.1007/s11095-013-1232-z, (accessed on 7 August 2022); ** US Environmental Protection Agency. Comptox Chemicals Dashboard. https://comptox.epa.gov/dashboard/chemical/details/DTXSID3037208 (accessed on 7 August 2022); ^§^ https://chemaxon.com/products/calculators-and-predictors#pka, (accessed on 7 August 2022); ° Computed by XLogP3 3.0 (PubChem release 7 May 2021); ^φ^ Sangster J; LOGKOW Databank. Sangster Res. Lab., Montreal Quebec, Canada (1994).

## Data Availability

Not applicable.

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
