# Peer review of "Combination-Based Strategies for the Treatment of Actinic Keratoses with Photodynamic Therapy: An Evidence-Based Review"

_pharmaceutics, 2022, doi:10.3390/pharmaceutics14081726_

Round 1

Reviewer 1 Report

Minor remarks:

1.      A comma is lacking after the name Mazzetto.

2.      Please improve language style in the abstract: “In this study, we reviewed the combination-based PDT treatments described in the literature for treating AK. Combination-based PDT treatments (post-treatment….” – there is no need in a repetition.

3.      Abbreviation YAG is not introduced in the abstract (unfortunately, there is no line numbering).

4.      List of abbreviations is not full there are no TMFI, YAG etc. It is true that they are specified in the text, but there should be consistency – or all the abbreviations are mentioned in the list, or all the abbreviations are explained in the text.

5.      The title of the Section 3 sounds very strange: “3. Systemic treatment combined with PDT Topical treatments combined with PDT”  - please correct it.

6.      In Section 4.1 the phrase “Thus, lasers can be used during pretreatment to facilitate the enrichment of ALA or MAL in dysplastic cells, and this approach is also known as laser-assisted drug delivery” is divided into 2 parts – please connect them.

7.      In Section 4.1 the lines “Er:YAG (erbium:yttrium-aluminium-garnet) laser” and “CO2 (carbon dioxide) laser” are actually sub-section titles – please number them.

8.      In Section 4.1 – “…. diode lamp at 37 J/cm, and the patients…” – are you sure that it is 37 J/cm and not 37 J/cm2 ? Please prove it.

Author Response

Reviewer 1 - Minor remarks:

  1. A comma is lacking after the name Mazzetto.

Re: We thank the Reviewer for noticing this omission. We added the comma after the name Mazzetto.

  1. Please improve the language style in the abstract: “In this study, we reviewed the combination-based PDT treatments described in the literature for treating AK. Combination-based PDT treatments (post-treatment….” – there is no need in repetition.

Re: We thank the Reviewer for the suggestion. We modified the text accordingly.

  1. Abbreviation YAG is not introduced in the abstract (unfortunately, there is no line numbering).

Re: We thank the Reviewer for the comment. We have now defined the abbreviation YAG in the abstract as requested.

  1. List of abbreviations is not full there are no TMFI, YAG etc. It is true that they are specified in the text, but there should be consistency – or all the abbreviations are mentioned in the list, or all the abbreviations are explained in the text.

Re: We now added a full list of abbreviations.

  1. The title of the Section 3 sounds very strange: “3. Systemic treatment combined with PDT Topical treatments combined with PDT”  - please correct it.

Re: We thank the reviewer for pointing this out. We have now corrected the text.

  1. In Section 4.1 the phrase “Thus, lasers can be used during pretreatment to facilitate the enrichment of ALA or MAL in dysplastic cells, and this approach is also known as laser-assisted drug delivery” is divided into 2 parts – please connect them.

Re: We thank the Reviewer for the suggestion. We modified the text accordingly.

  1. In Section 4.1 the lines “Er:YAG (erbium:yttrium-aluminium-garnet) laser” and “CO2(carbon dioxide) laser” are actually sub-section titles – please number them.

Re: We thank the Reviewer for the suggestion. We modified the sub-section number as requested.

  1. In Section 4.1 – “…. diode lamp at 37 J/cm, and the patients…” – are you sure that it is 37 J/cm and not 37 J/cm2? Please prove it

Re: We thank the reviewer for pointing out this typographical error. It has been corrected.

Reviewer 2 Report

1.       I think the current title “Combination-based or pretreatment strategies for the treatment of actinic keratoses with photodynamic therapy: a review” should be changed in a manner to be more informative. It may be changed to “Combination-based strategies for the treatment of actinic keratoses with photodynamic therapy: an evidence-based review”.

2.       English editing by a native editor can significantly help authors for a better scientific writing.

3.       The abstract’s results section should be critically revised. It is confusing. Options should be categorized based on their different groups.

4.       PDT should be removed from the keywords. Please check the spelling of actinic keratosis (a uniformed spelling should be used through the manuscript).

5.       This subsection (Systemic treatment combined with PDT Topical treatments combined with PDT) should be revised.

6.       The section “4.1. Laser-assisted PDT” must have several numbered subsections.

7.       This reference (Y Harth; B Hirshowitz; B Kaplan. Modified topical photodynamic therapy of superficial skin tumors, utilizing aminolevulinic acid, penetration enhancers, red light, and hyperthermia. Dermatol Surg 1998 24(7):723-6; DOI: 10.1111/j.1524-4725.1998.tb04240.x) has no number!

8.       Table 1 is incompletely shown in the PDF file. Data under the column “OUTCOME OF SEQUENTIAL REGIMEN vs PDT ALONE” must be well-organized to be more understandable. The significance of each different treatment must be mentioned.

9.       The conclusion section is not suitable. It is very long and boring and not conclusive.

Author Response

  1. I think the current title “Combination-based or pretreatment strategies for the treatment of actinic keratoses with photodynamic therapy: a review” should be changed in a manner to be more informative. It may be changed to “Combination-based strategies for the treatment of actinic keratoses with photodynamic therapy: an evidence-based review”.

Re: We thank the Reviewer for the suggestion and changed the title accordingly.

  1. English editing by a native editor can significantly help authors for a better scientific writing.

Re: We thank the reviewer for the observation. The text was revised by a native editor.

  1. The abstract’s results section should be critically revised. It is confusing. Options should be categorized based on their different groups.

Re:  We thank the Reviewer for pointing this out. We now categorized different options in groups and revised the section for readability and format.

  1. PDT should be removed from the keywords. Please check the spelling of actinic keratosis (a uniform spelling should be used throughout the manuscript).

Re: We thank the author for the suggestion. We now used only “actinic keratoses”. However, we prefer not to remove “PDT“ from the list of keywords since photodynamic therapy is commonly defined PDT and several readers use “PDT” as a keyword when searching the literature.

  1. This subsection (Systemic treatment combined with PDT Topical treatments combined with PDT) should be revised.

Re: We thank the Reviewer for pointing out this mistake. We corrected the typographical error.

  1. The section “1. Laser-assisted PDT” must have several numbered subsections.

Re: We have now added sub-section number as suggested.

  1. This reference (Y Harth; B Hirshowitz; B Kaplan. Modified topical photodynamic therapy of superficial skin tumors, utilizing aminolevulinic acid, penetration enhancers, red light, and hyperthermia. Dermatol Surg 1998 24(7):723-6; DOI: 10.1111/j.1524-4725.1998.tb04240.x) has no number!

Re: We thank the Reviewer for pointing this out. We have numbered the reference as recommended.

  1. Table 1 is incompletely shown in the PDF file. Data under the column “OUTCOME OF SEQUENTIAL REGIMEN vs PDT ALONE” must be well-organized to be more understandable. The significance of each different treatment must be mentioned.

Re: We thank the Reviewer for the comment. We modified the table for better clarity and added data on the significance of the differences between treatments.

  1. The conclusion section is not suitable. It is very long and boring and not conclusive.

Re: Thanks to the reviewer's advice. We have summarized the conclusion section.

Reviewer 3 Report

The current manuscript "Combination-based or pretreatment strategies for the treatment of actinic keratoses with photodynamic therapy: a review" mainly focus on the combination of therapy for the treatment of actinic keratoses. after reviewing the manuscript i recommend major revison.

Following are the major concerns

1. No figure in the whole manuscript. Authors should add/draw figures for easy understanding of readers. 

2. Add more and latest references. 

3. Add some statistics, comparision between different treatment groups. 

4. which treatment group is best for the actinic keratoses and why? should be clearly mentioned in the manuscript.

5. Auothors used frequently the words "we and our" in the manuscript. try to avoid frequent use of these words. 

Thanks and Regards

Author Response

Following are the major concerns

  1. No figure in the whole manuscript. Authors should add/draw figures for easy understanding of readers. 

Re: We thank the Reviewer for the comment. Accordingly, we have added 2 figures to illustrate PDT mechanism of action and each combination-based strategies for the treatment of actinic keratoses with PDT.

  1. Add more and latest references. 

Re: We thank the Reviewer for the suggestion. We have added recent references.

  1. Add some statistics, comparison between different treatment groups. 

Re: Unfortunately, we couldn’t perform any comparison between different studies due to the heterogeneity among them. Nonetheless, we completed table 1 with p-values to help the reader to compare different therapies.

  1. which treatment group is best for the actinic keratoses and why? should be clearly mentioned in the manuscript. CONCLUSION

Re: We thank the Reviewer for the suggestion. We have now specified the most effective combination treatments for treating AKs and the best choices for different clinical scenarios (immunosuppressed patients, acral areas, and thicker AKs).

  1. Authors used frequently the words "we and our" in the manuscript. Try to avoid frequent use of these words. 

Re: We thank the reviewer for the comment, and we amended some sentences and corrected some parts of the article as suggested.

Reviewer 4 Report

Dear Authors,

A comprehensive review. This is an important area and a well written review was needed.

 However, there’s a few things that can be addressed.

·         The only table in the manuscript is not presented properly. Last column can’t be seen.

·         Would be good if more information on the molecules could be added to the table. Especially, logP and Pka

·         Recent literature can be sighted (https://doi.org/10.1016/j.addr.2022.114293) and (https://doi.org/10.1016/j.addr.2021.113929)

·         Would be valuable to add comments on the skin penetration of the combination molecules. Most of them are hard to penetrate the skin under normal conditions.

Author Response

  1. The only table in the manuscript is not presented properly. Last column can’t be seen.

Re: We thank the Reviewer for pointing this out. We have now modified the layout of the table.

  1. Would be good if more information on the molecules could be added to the table. Especially, logP and Pka

Re: We thank the reviewer for the suggestion. We have now included logP and Pka in the table.

  1. Recent literature can be sighted (https://doi.org/10.1016/j.addr.2022.114293) and (https://doi.org/10.1016/j.addr.2021.113929)

Re: Thank you for the comment. Some recent articles (including those suggested) have been now included.

  1. Would be valuable to add comments on the skin penetration of the combination molecules. Most of them are hard to penetrate the skin under normal conditions.

Re: We thank the reviewer for the suggestion. Most of the topical drugs mentioned in our review act as differentiation-promoting agents, enhancing PpIX production in tumor cells, instead of increasing the penetration of MAL/ALA. Therefore, we believe that a comment on the skin penetration of the combination molecules may not be essential.

Round 2

Reviewer 3 Report

Dear Editor,

Hope you will be doing well.

The authors have addressed all the comments in the manuscript. Therefore, I recommend this manuscript for publication.

Thanks and Regards

This manuscript is a resubmission of an earlier submission. The following is a list of the peer review reports and author responses from that submission.

Round 1

Reviewer 1 Report

This is an interesting review regarding the combination of Photodynamic therapy (PDT) with other therapeutic modalities for the treatment of actinic keratosis. Particularly the review is focused in the modulation of PDT response after pretreatment of actinic keratosis with specific topical, systemic or mechanical-physical strategies.  

The review is wery well organized and well described and I consider that  it is important for researches involved in the treatment of actinic keratosis with PDT.

Reviewer 2 Report

In this review the authors are concerned with how to improve ALA PDT for AK  by combination with pre-treatments or post-treatments. This is a timely and useful review. I have the following remarks:

  1.  It would be useful for the non-dermatologist readers of Pharmaceutics to have a list explaining the abbreviations you use at the beginning of the paper.
  2. There seem to be some possibilities for enhancing PDT of AK to be missing. For instance penetration enhancers like DMSO or putting the ALA in particular creams could be mentioned, as well as low frequency ultrasound, or even iontophoresis in the case of Metvix (see publication by R. Gurny et. al.). Chemical peeling with glycolic acid, using sticky tape to strip off the stratum corneum is actually simple and used a lot, and the addition of deferral (desferrrioxamine). Some of these actually work well and should probably be added for making the review more complete. More recently the enhanced ALA-based PDT by photobiomodulation was published by J. Joniova et. al. in J. Photochemical + Photobiol. B, Biology, 2021, 225, 112347. 
  3. It would be useful in the introduction to mention the other treatments, i.e. the non-PDT treatments like cryo-therapy, which work quite well and are easy to apply, and discuss advantages/disadvantages.
  4. When talking about ALA-based PDT it may also be useful to briefly discuss certain advantages like the fact that scarring tends to be much less in the case of PDT as the connective tissue tends to be left in place (only cells are destroyed). Significant disadvantages include pain. 
  5. Concerning PDT-induced pain, it could be mentioned that low light intensity, i.e. longer irradiation times (as would be the case in daylight PDT for instance) tend to reduce pain.
  6. As far as light parameters are concerned there is little precise information in your review. i.e. When should red light be used and when blue light, possibly depending on the thickness of the lesion. Please add some information on this.
  7. A useful reference that you may want to add is: D. B. Eisen et. al. in the Journal of the American Academy of Dermatology  Oct. 2021, 85(4): "Guidelines of care for the management of actinic keratosis".
  8. It is not clear if immunosuppressed patients are included in the review or not (see lines 476 and 68 which seem contradictory).
  9. The main outcomes you discuss tend to be related to the number of AKs that have disappeared. Is there any information on the progression to squamous cell carcinoma?
  10. Tirbanobulin ointment is not mentioned (microtubili inhibitor which works against AK). Any reason for this?

Reviewer 3 Report

The m/s pharmaceutics-1729697 “Combination-based or pretreatment strategies for the treatment of actinic keratoses with photodynamic therapy: a review” deal with a very important issue of combating keratoses, but despite the importance of the topic, the review cannot be accepted in the present form.

  1. Review is not supposed to conclude the sections of Methods, Results and Discussion (also in the abstract). There is no need to write about the sources of search.
  2. The Introduction is very shallow and does not contain relevant information.
  3. Sections 2.1 and 2.2 do not relate to Methods
  4. All the review should be re-structured and divided into logic sections.